# Teratogenic, Oxidative Stress and Behavioural Outcomes of Three Fungicides of Natural Origin (*Equisetum arvense*, *Mimosa tenuiflora*, Thymol) on Zebrafish (*Danio rerio*)

**DOI:** 10.3390/toxics9010008

**Published:** 2021-01-09

**Authors:** Raquel Vieira, Carlos Venâncio, Luís Félix

**Affiliations:** 1Centre for the Research and Technology of Agro-Environmental and Biological Sciences (CITAB), University of Trás-os-Montes and Alto Douro (UTAD), 5000-801 Vila Real, Portugal; raquelvieira1920@gmail.com (R.V.); cvenanci@utad.pt (C.V.); 2Department of Animal Science, School of Agrarian and Veterinary Sciences (ECAV), University of Trás-os-Montes and Alto Douro (UTAD), 5000-801 Vila Real, Portugal; 3i3S-Instituto de Investigação e Inovação em Saúde, Universidade do Porto, 4200-135 Porto, Portugal; 4Laboratory Animal Science, IBMC-Instituto de Biologia Molecular Celular, Universidade do Porto, 4200-135 Porto, Portugal

**Keywords:** natural products, fungicides, early development, teratogenicity, zebrafish, behaviour, oxidative stress

## Abstract

The improper use of synthetic fungicides has raised public concerns related to environmental pollution and animal health. Over the years, plant-derived antifungals have been investigated as safer alternatives, although little scientific evidence of its neurodevelopmental effects exist. The main objective of this study was to explore the effects of three alternative natural extracts (*Equisetum arvense*, *Mimosa tenuiflora*, Thymol) with antifungal properties during the early development of zebrafish by evaluating different teratogenic, oxidative stress and behavioural outcomes. Following the determination of the 96 h-LC_50_, exposure to sublethal concentrations showed the safety profile of both *E. arvense* and *M. tenuiflora*. However, following 96-h exposure to Thymol, increased lethality, pericardial oedema, yolk and eye deformations, and decreased body length were observed. The reduced and oxidized glutathione (GSH:GSSG) ratio was increased, and the glutathione-s-transferase activity in the group exposed to the highest Thymol concentration. Overall, these results support a more reducing environment associated with possible effects at the cellular proliferation level. In addition, the disruption of behavioural states (fear- and anxiety-like disorders) were noted, pointing to alterations in the c-Jun N-terminal kinase developmental signalling pathway, although further studies are required to explore this rationale. Notwithstanding, the results provide direct evidence of the teratogenic effects of Thymol, which might have consequences for non-target species.

## 1. Introduction

The use of agrochemicals to control plant diseases has become crucial in modern agricultural procedures, with a diverse range of commercial products being released over the last years [1]. The improper application and extensive use of these compounds have raised public concerns related to environmental pollution and animal health [2]. In fact, agrochemical residues spread in aquatic systems [2,3], compromising not only aquatic food resources but also fisheries and aquaculture. However, the effects of fungicides in non-target species have received less attention when compared to herbicides and insecticides [4]. In this context, although acceptable regulatory concentrations have been established for pesticide residues [5], local and regional studies have documented the worldwide occurrence of synthetic fungicides in surface waters in concentrations up to around 80 μg L^−1^, which are generally higher than those observed for herbicides and insecticides (reviewed by [4]). Furthermore, these concentrations are superior to the acceptable regulatory concentrations and above the average lowest effective concentrations in different non-target aquatic biota (reviewed by [4]).

Over the past years, phyto-fungicides (plant-based extracts or compounds) have been investigated as an alternative to synthetic fungicides and commercialised for the management of a wide range of fungal diseases in plants [6,7,8], and they are generally accepted as safe, easily biodegradable, environmentally friendly and with low toxicity [9,10]. However, their use is often limited due to their instability and rapid degradation, requiring higher application rates and application frequency [11]. In addition, adverse or toxic side effects for non-target species are usually reported on labels or material safety data sheets, but there is a paucity of scientific and ecotoxicological information, in particular, during neurodevelopmental periods, which are known to be critical for the population dynamics and ecosystem functioning [12]. For instance, horsetail (*Equisetum arvense*) was the first approved basic substance according to the European Regulation (EC) 1107/2009 [13], but, although not considered as a substance of concern, it has been associated with potential neurodevelopmental toxicity [14]. Likewise, “jurema preta” (*Mimosa tenuiflora*) has been described as teratogenic to higher vertebrates [15,16], and Thymol, the main monoterpene phenol isolated from plants from the *Lamiaceae* family such as *Thymus vulgaris* L. [17], has been shown to cause developmental abnormalities in chicken and zebrafish embryos [18,19,20] and to regulate cholinergic and antioxidant systems in cognitive dysfunctional zebrafish [21]. However, although current research shows its antifungal properties [17,22,23] and some commercial pesticide products based on natural compounds are available on the market, there is a lack of sufficient neurodevelopmental information, and further studies are needed to clarify their environmental risk to non-target organisms to define efficient and appropriate use patterns.

Among aquatic organisms, fish have been considered useful biological indicators for ecotoxicological studies [24], with the powerful and versatile teleost vertebrate zebrafish (*Danio rerio*) model being increasingly used [25,26]. In fact, its low husbandry costs, reproduction potential and embryonic external fertilization method, rapid development and transparency as well as the low ethical constrains associated with considering its embryonic stages facilitate high-throughput screenings [27,28]. Furthermore, the literature supports zebrafish embryos as more responsive to test compounds in comparison to adult fish [29,30], and they are a conservative indication of later biological changes [31]. Therefore, considering these characteristics, the present study was focused on the toxicological effects of commercially available herbal products containing *Equisetum arvense*, *Mimosa tenuiflora* and Thymol on the zebrafish embryo. In particular, the aims of this study were to evaluate the (1) morphological and physiological changes, (2) oxidative status alterations and (3) behavioural impacts induced by the test formulations during early zebrafish development.

## 2. Materials and Methods

### 2.1. Chemicals and Reagents

The commercial formulation of horsetail extract (*Equisetum arvense*) decoction (95.2% decoction of horsetail (*E. arvense* 7%), tansy herbs (*Tanaceti Hb*. conc. 2.5%), wormwood (*Artemisia absinthium* 1%) and 4.8% humus extract (15% wine marc extract)) was acquired from Aries Umweltprodukte (Horstedt, Germany). The ethanolic extract of *Mimosa tenuiflora* (Matry, 80% M. tenuiflora extract containing 1% zinc and 1% manganese) was purchased from Biagro (Valencia, Spain) and Thymol (extra pure, CAS 89-83-8) was acquired from EMD Millipore (Oeiras, Portugal). Based on the percentage of the principal component (*E. arvense*, *M. tenuiflora*, and Thymol) stock solutions of 6250, 80,000, and 500 mg L^−1^ were prepared for the *E. arvense*, *M. tenuiflora* and Thymol, respectively, and stored at 4 °C. Exposure solutions were freshly prepared in embryo water (28.0 ± 0.5 °C, 200 mg L^−1^ Instant Ocean Salt and 100 mg L^−1^ sodium bicarbonate; UV sterilized) prepared from City of Vila Real filtered tap water. Except when specified, all other chemicals were of the highest grade commercially available and obtained from standard commercial suppliers.

### 2.2. Animals

Adult wild-type AB strain zebrafish (*Danio rerio*) were maintained at the University of Trás-os-Montes and Alto Douro (Vila Real, Portugal). Fish were maintained under standard conditions at 28.0 ± 0.5 °C with a 14 h light/10 h dark photoperiod in an open water system with continuous supply of aerated, dechlorinated, charcoal-filtered and UV-sterilized City of Vila Real tap water (pH 7.3–7.5). Fish were fed with a commercial diet (Sera, Heinsberg, Germany) supplemented with *Artemia* sp. *nauplii* twice a day. Embryos were collected by the natural spawning method by maintaining a 2:1 male to female ratio in cages overnight. Embryos were collected within 1 h after the onset of the light cycle (at 8.00 a.m.), rinsed, and bleached with a diluted Chloramine-T solution (0.5% *w*/*v*), washed twice with embryo water and transferred into a petri dish for egg selection. Embryonic stages were denoted as hours post-fertilization (hpf) under a SMZ 445 stereomicroscope (Nikon, Japan) and 2 hpf normal fertilized embryos were used for the subsequent experiments. All animal procedures were performed in accordance with the ethical principles and other requirements on the use of laboratory animals of the EU directive (2010/63/EU) and national legislation for animal experimentation and welfare (Decreto-Lei 113/2013). In addition, two authors have a level B FELASA certification (Federation of European Laboratory Animal Science Associations) while another author has a FELASA C certification.

### 2.3. Concentration Determination

The OECD testing guideline 236 was applied to determine the lethal concentration that causes 50% mortality (LC_50_) with modifications. Embryos at 2 hpf were randomly distributed in 6-well culture plates (5 mL solution and 20 embryos per well) for triplicate exposure to seven test solutions (10-fold dilution from the stock solutions) for 96 h under the controlled conditions reported before. Embryo water was used as blank control and to prepare all test solutions. The exposure solutions were renewed every 24 h to keep the appropriate concentrations and water quality. The embryonic mortality was recorded daily and following correction for the percentage of mortality in the control group using Abbott’s formula, and the 96 h-LC_50_ values were determined using the probit analysis. The 96 h-LC_50_ and the 95% confidence limits were calculated as 1.98 mg L^−1^ (0.50–4.13), 1.55 mg L^−1^ (0.39–3.44), and 2.35 mg L^−1^ (0.78–5.55) (Appendix A), respectively, for *E. arvense*, *M. tenuiflora*, and Thymol. Based on the calculated LC_50_, three sub-lethal concentrations were selected for the subsequent experiments.

### 2.4. Embryo Toxicity

Based on the LC_50_ calculation, 0.00625—E1, 0.0625—E2 and 0.625 mg L^−1^—E3 (about 1/300, 1/30 and 1/3 of the LC_50_) were chosen for the *E. arvense* based formulation. For the *M. tenuiflora*, the selected concentrations were 0.008—M1, 0.08—M2 and 0.8 mg L^−1^—M3 (around 1/200, 1/20 and 1/2 of the LC_50_) while for Thymol the concentrations for testing were 0.01—T1, 0.1—T2 and 1 mg L^−1^—T3 (approximately 1/200, 1/20 and 1/2 of the LC_50_). Normally developed 3 h post fertilization (hpf) fertilized eggs were randomly placed in 6-well culture plates (50 embryos in 5 mL solution/well) and exposed to the above solutions. A blank control group (embryo water only) was also prepared and included in each plate. The plates were maintained in a constant temperature-light cycle (28 °C and 14:10 h light-dark cycle) for a period of 96 h, after which eleutheroembryo were washed twice and allowed to develop until 120 hpf (Figure 1). During the experimental period, the exposure solutions were replaced daily to maintain the appropriate concentration of the test compounds. The experiments were repeated independently five times. The zebrafish development (10 random animals removed from each group) was accompanied under a SMZ800 stereomicroscope with the cumulative mortality being assessed at 8, 24, 48, 72 and 98 hpf according to the standard guidelines [32], with dead embryos removed from the plates. Lethal parameters such as failure of somites, eye and otolith development, missing heartbeat, and nondetached tail and head, were recorded at 24, 48, 72, and 98 hpf according to previous studies [33,34]. The spontaneous movements at 24 hpf, pigmentation formation and heart rate at 48 hpf and hatching rate at 72 hpf were evaluated as sublethal endpoints. Morphological abnormalities (body length, area of egg yolk, area of heart and eye, and head to body angle) were screened at 98 hpf in 10 randomly 3% methylcellulose-immobilized eleutheroembryo. Images of morphological defects were photographed using an inverted microscope (IX 51, Olympus, Antwerp, Belgium) and combined, merged, and processed with Adobe Photoshop CS6 (Adobe Systems, San Jose, CA, USA). Measurements were taken using the Digimizer software (version 4.1.1.0, MedCalc Software, Mariakerke, Belgium). Eleutheroembryo were further collected for subsequent biochemical analysis or washed three times with embryo medium and allowed to develop until 120 hpf for behavioural analysis. In total, five independent replicates from independent spawns were used to maximize the genetic variability of the individuals.

### 2.5. Biochemical Analysis

After exposure to the test compounds for 96 h, biochemical analysis was conducted as detailed before [35]. Around 30 randomly selected eleutheroembryo from each group were homogenized in 400 µL cold buffer (0.32 mM of sucrose, 20 mM of HEPES, 1 mM of MgCl_2_, and 0.5 mM of phenylmethyl sulfonylfluoride (PMSF), pH 7.4) in a Tissuelyser II (30 Hz for 30 s, Qiagen, Hilden, Germany). Following a centrifugation at 12,000× *g* for 10 min at 4 °C in a refrigerated centrifuge (Sigma 3K30, Osterode, Germany), supernatant protein concentration was determined using the Bradford method at 595 nm with bovine serum albumin (BSA) as a standard. The overall reactive oxygen species (ROS) generation was measured using 2′,7′-dichlorofluorescein diacetate (DCFH-DA) at 485 nm (excitation) and 530 nm (emission). Changes in oxidative stress indicators, such as the activity of superoxide dismutase (Cu/Zn-SOD) and catalase (CAT), were evaluated at 560 nm and at 240 nm, respectively. The activity of glutathione peroxidase (GPx) and glutathione-s-transferase (GST) were measured at 340 nm. The reduced (GSH) and oxidized glutathione (GSSG) were derivatized with ortho-phthalaldehyde (OPA) and measured at 320 nm and 420 nm for excitation and emission wavelengths, respectively. The ratio between glutathione (GSH:GSSG) was used to describe the redox ratio (oxidative stress index, OSI). The content of malondialdehyde (MDA) was estimated by the quantification of the MDA-TBA adducts at 530 nm with a correction for non-specific adducts at 600 nm. The lactate dehydrogenase (LDH) was assayed at 340 nm and the acetylcholinesterase (AChE) activity at 405 nm. All samples (10 µL) were tested in duplicate and measured against a reagent blank at 30 °C using a PowerWave XS2 microplate scanning spectrophotometer (Bio-Tek Instruments, Winooski, VT, USA) or a Varian Cary Eclipse (Varian, Australia) spectrofluorometer, equipped with a microplate reader. To integrate all the biomarker responses into a general “stress index”, the integrated biomarker response index version 2 (IBRv2) was calculated according to a previous method [36] representing the stress level at each tested concentration, based on the principle of reference deviation. Overall, data were normalized to control values and log-transformed (Yi) to diminish variability, and the overall mean (μ) and standard deviation (s) calculated. Data were further standardized as Zi = (Yi − μ)/s. The difference between Zi and Z0 (control) was then calculated to determine A values and the IBRv2 was calculated by summing the absolute values of A.

### 2.6. Locomotor Behaviour Analysis

The zebrafish eleutheroembryo locomotor behaviour (exploratory open field test), the patterns of avoidance (in response to a bouncing ball stimulus) and anxiety-like behaviours (in the visual motor response test) were analysed 24 h after the end of the exposure, at 120 hpf, in a climatized dark room as previously described [34,35,37]. Briefly, 6-well agarose-coated plates containing 1 randomly picked eleutheroembryo per well (5 per group) were placed above a 15.6″ laptop LCD screen (1366 × 768 pixels resolution) showing a white Microsoft PowerPoint (Microsoft Corp., Washington, DC, USA) presentation. A 14.2 megapixels Sony Nex-5 digital camera was used to record the exploratory behaviour (mean speed, total distance moved, percentage of time spent in each zone, mean distance to centre zone (5 mm radius circle) of the well, mean absolute turn angle, and percentage of time active) of the eleutheroembryo during 10 min after a period of acclimation (5 min). After the analysis of exploratory behaviour, the avoidance response was measured by the eleutheroembryo’s ability to respond to a visual stimulus (a red bouncing ball present at the upper half of the well and moving from left to right) provided by the presentation in the Microsoft PowerPoint (Microsoft Corp., Redmond, WA, USA) during alternating periods (10 min). In addition, the anxiety-like behaviour of eleutheroembryo was monitored in duplicate conditions of continuous visible light (10 min) and dark (10 min) using an infrared-capable camera (GENIUSPY, GS-NQ140CML) with a 3.6 mm lenses using the same plate configuration. The TheRealFishTracker software was used to video-track individuals and eleutheroembryo exhibiting obvious malformations in the exposure, and control groups were excluded to avoid the interference of morphological effects.

### 2.7. Statistical Analysis

The statistical analyses were performed on the averaged values from each independent exposure using the GraphPad Prism software (Prism 8). The LC_50_ values were calculated using a variable slope model. The normality of data was controlled using Shapiro Wilk’s test, and the homoscedasticity was checked with Brown-Forsythe’s test. When data followed the normal distribution, differences among groups were assessed by one-way analysis of variance (ANOVA) followed by the Tukey multiple comparison test and data expressed as mean ± standard deviation (SD). When data followed a non-normal distribution, the data treatment was performed using the non-parametric Kruskal-Wallis analysis of variance followed by Dunn’s test with a Bonferroni correction for multiple comparisons and data expressed as medians and interquartile range (25th; 75th percentiles). The student’s *t* test was used to evaluate differences for the aversive behavioural responses. A *p* < 0.05 was considered to be a statistically significant difference.

## 3. Results

### 3.1. Teratogenic Effects of Phyto-Fungicide Formulations

The effects on the embryo development were evaluated from ~2 hpf and for a period of 96 h with different parameters being evaluated at specific time-points (Table 1). At 24 hpf, no significant changes were observed in the development of the tail, head and somites following exposure to any of the phyto-fungicides (Appendix A). Similarly, the spontaneous movements were not affected by *E. arvense* (X^2^(3) = 7.859, *p* = 0.050) or *M. tenuiflora* (X^2^(3) = 1.037, *p* = 0.792), nor by Thymol (X^2^(3) = 4.907, *p* = 0.179) (Table 1). At 48 hpf, the eyes, otoliths, pigmentation, and blood circulation were visible in all treated embryos (Appendix A), and no changes were depicted in the heart rate of the individuals for *E. arvense* (F(3,16) = 3.065, *p* = 0.058) or *M. tenuiflora* (F(3,16) = 0.649, *p* = 0.595), nor by Thymol (F(3,16) = 2.357, *p* = 0.110) (Table 1). At 72 hpf, and despite slight variations in the Thymol-exposed individuals (X^2^(3) = 2.159, *p* = 0.540, Appendix A), no significant changes were apparent for the oedema presence. At this time-point, as shown in Table 1, the hatching rate did not differ among *E. arvense* (X^2^(3) = 2.201, *p* = 0.532), *M. tenuiflora* (F(3,15) = 1.039, *p* = 0.404) or Thymol (F(3,16) = 0.145, *p* = 0.931) treated embryos.

At 98 hpf, embryo development in the control groups was as expected with around 80% of the animals showing a normal development with mortalities of about 10%, and 6 to 10% malformed individuals (Figure 2A) without significant changes after 96 h exposure to *E. arvense* or to *M. tenuiflora* (*p* > 0.05). However, after 96 h exposure to Thymol (X^2^(3) = 12.46, *p* = 0.006), the cumulative mortality increased significantly after exposure to T3 (*p* = 0.004) in relation to the control group, while no significant differences were verified between the other groups. Similarly, at this time point, malformed eleutheroembryos (X^2^(3) = 9.827, *p* = 0.020) were noticed in T3 group (*p* = 0.029), showing a higher percentage in relation to the control group (Figure 2A,B). The quantitative analysis (Appendix A) showed that the most evident malformations were related to the yolk (X^2^(3) = 11.81, *p* = 0.008), pericardial (F(3,15) = 3.516, *p* = 0.041), and eye (X^2^(3) = 9.377, *p* = 0.025) areas and to the overall body length of the eleutheroembryo (F(3,15) = 6.231, *p* = 0.006). In this regard, exposure for 96 h to Thymol caused a decreased yolk (*p* = 0.034 between T3 and the control group and *p* = 0.017 between T3 and T1), an increased pericardial (*p* = 0.042 between T3 and the control group), and a decreased eye (*p* = 0.039 between T3 and the control group). In addition, exposure to T2 and T3 caused a significant reduction on the body length of 98 h eleutheroembryos in relation to the control group (*p* = 0.026 and *p* = 0.018, respectively).

### 3.2. Biochemical Markers Affected by the Phyto-Fungicide Exposure

The biochemical changes induced by the exposure to these phyto-fungicides were evaluated at the end of the exposure by some ROS-mediated and related parameters, which were normalised to control values and are summarized in Figure 3 (original data is shown in Appendix A). After exposure to E1, an elevated AChE activity (F(3,16) = 3.526, *p* = 0.039) was detected in relation to the control group (*p* = 0.041). No other difference was noted after exposure to *E. arvense*. Similarly, no biochemical changes were observed in zebrafish following exposure to *M. tenuiflora*. However, when embryos were exposed to Thymol, an increase in the GSH:GSSG ratio (F(3,16) = 11.21, *p* < 0.001) was observed for T1 (*p* < 0.001), T2 (*p* < 0.001) and T3 (*p* = 0.025) in relation to the control group. Exposure to T3 also resulted in an increased activity of GST activity (X^2^(3) = 12.60, *p* = 0.006) in relation to the control group (*p* = 0.003). No other change was perceived. The star plot representations for each compound (Figure 3B) shows how each individual biomarker contributed to the IBRv2 index obtained for each compound. Overall, a negative relationship between the IBRv2 values for *E. arvense* and *M. tenuiflora* was obtained with the lowest concentrations showing higher values in relation to the lowest concentrations which may be associated to the individual changes observed. On the other hand, the IBRv2 index was similar in the Thymol exposed groups although changes were observed in the individual biomarkers.

### 3.3. Behavioural Responses Induced by the Different Formulations

The behavioural responses evaluated at 120 hpf after the 96-h exposure to the phyto-fungicides are shown in Figure 4. Regarding the exploratory behaviours, no significant changes were observed following exposure (Figure 4A and Appendix A). Concerning the ability to escape the red bouncing ball (aversive stimulus, Figure 4B), the individuals exposed to *E. arvense* and *M. tenuiflora* showed their ability to escape from the stimulus by remaining for more time in the area without the stimulus (*p* < 0.05). However, after exposure to Thymol, eleutheroembryo from the T2 and T3 group showed a reduced ability to escape the aversive stimulus (*p* = 0.453 and *p* = 0.765, respectively), spending the same amount of time in both halves of the well. The individuals were also tested for anxiety-like behavioural changes using the light/dark test and the results are shown in Figure 4C. In comparison to the control group, no significant changes were perceived after exposure to *E. arvense* and *M. tenuiflora* regardless of the lightning conditions (for *E. arvense*: 10 min: F(3,16) = 3.325, *p* = 0.050; 20 min: F(3,16) = 8.286, *p* = 0.002 with significant differences between E1 and E3 (*p* = 0.002) and between E2 and E3 (*p* = 0.014); 30 min: F(3,15) = 1.370, *p* = 0.290 and 40 min: F(3,16) = 0.578, *p* = 0.638 and for *M. tenuiflora*: 10 min: F(3,14) = 3.469, *p* = 0.045 with significant differences between M1 and M3 (*p* = 0.030); 20 min: F(3,16) = 0.736, *p* = 0.546; 30 min: F(3,15) = 2.129, *p* = 0.155 and 40 min: F(3,16) = 0.510, *p* = 0.682). Thymol exposure during initial zebrafish development induced no changes on the first light period (F(3,16) = 0.458, *p* = 0.715), but exposure to T3 induced hyperactivity in relation to the control group (*p* = 0.047) in the first dark period (F(3,16) = 2.919, *p* = 0.023). However, these differences disappeared in the second light (F(3,14) = 1.602, *p* = 0.234) and dark (F(3,16) = 3.223, *p* = 0.061) periods.

## 4. Discussion

The worldwide occurrence of synthetic fungicides in aquatic environments has led to the investigation of different plant-based extracts or compounds as safer alternatives with low toxicity. However, adverse or toxic side effects have been reported in higher vertebrates [14,15,16,18]. Furthermore, there is a lack of ecotoxicological information about these products, which can have environmental implications. In this study, the toxicological effects of commercially available herbal products containing *E. arvense*, *M. tenuiflora*, and Thymol were tested using zebrafish embryos. The results showed the safety of *E. arvense* and *M. tenuiflora* with no lethality, and only a slight increase of AChE activity was observed following exposure to the lowest concentration of *E. arvense*, which may have no biological significance as no other association with both biochemical and behavioural markers could be made despite the IBRv2 index suggesting higher stress levels in this concentration. On the other hand, after exposure to Thymol, mortality and malformed development were observed. In addition, changes in the glutathione ratio and the disruption of behavioural responses were perceived, although no changes in the IBRv2 index were depicted.

Results from the current study highlighted a higher toxicity of Thymol to zebrafish in early life stages in comparison to the remaining test compounds. While the calculated LC_50_ values are in line with those observed for both synthetic [35,38] and other natural compounds in this species (reviewed in [39]), no studies have been found in the literature regarding the embryo toxicological effects of *E. arvense* and *M. tenuiflora* in aquatic species, although a previous study has shown a higher EC_50_ for *E. arvense* extract in another aquatic model (*Daphnia* sp., 50–100 mg L^−1^ [40]). Additionally, a previous study has shown that Thymol exhibits a lower toxicity (3× higher LC_50_ of 42.35 μM~6.36 mg L^−1^) [19] in comparison to the current study, which could be explained by different species’ sensitivities among different laboratories. Notwithstanding, craniofacial and skeletal deformities were similarly observed at higher concentrations (50 μM~7.5 mg L^−1^ [19] and 40 mg L^−1^ [20]), further supporting the teratogenic potential of Thymol to zebrafish embryos, even at lower concentrations. During embryogenesis, cells acquire distinct functions and specific positions, giving rise to a functional, complex, and multicellular organism through a set of early molecular and cellular mechanisms [41]. The modulation of these signalling pathways is required for the early patterning decisions, and previous studies have shown that changes in these signalling pathways result in defective development [42,43]. Although information is limited, Thymol is known to play multiple modulatory roles. For instance, it has been shown to down-regulate PI3K/Akt and ERK pathways [44], which are the key mechanisms involved in cell growth, proliferation, differentiation and survival [45,46]. However, the complex interplay between these pathways, the knockdown of PI3K/Akt signalling genes have been associated with embryonic lethality (reviewed and summarised by [47]), which may justify the observed effects. In addition, the negative regulation of PI3K/Akt signalling by the overexpression of its inhibitor (PTEN) has been shown to impair cell movements during gastrulation, resulting in developmental defects, including heart oedema, small or missing eyes and short tail [48], as observed after Thymol exposure. Overall, these studies demonstrate that the PI3K/Akt signalling pathway may play an integral role in the teratogenicity of Thymol, although the underlying mechanism remains to be defined, requiring further insights.

Notwithstanding, previous studies have shown that the inhibition of PI3K/Akt signalling hinders the activation of Nrf2 [49], the master regulator of the anti-oxidative response. Changes in the Nrf2-mediated antioxidant response have been previously described in Thymol-induced malformations of zebrafish embryos [19]. This is a crucial antioxidant signalling molecule for developmental processes [50], and deficiencies in its levels have been shown to induce embryonic lethality and severe oxidative stress in mice [51]. Collectively, data gathered from previous studies point to the Nrf2-antioxidant signalling pathway, and its activation by oxidative stress plays a pivotal role in the teratogenesis of Thymol. Oxidative stress results from an imbalance between the production and accumulation of oxygen reactive species (ROS), which can impair embryonic development [52]. Among the different oxidative stress-related parameters, glutathione-associated assays are often the primary choice [53], with the calculation of the redox status (GSH:GSSG ratio) being traditionally reported as a biomarker of oxidative stress [53]. Although the dynamics of glutathione during early development are yet to be understood, changes in its levels are associated with developmental effects [54]. Yet, in the current study, no changes were observed for GSH and GSSG levels following exposure to Thymol. However, a significant increase in the GSH:GSSG ratio was observed. The interplay between glutathione expression and changes in the redox state is important for the correct development of the organism [55]. A higher GSH:GSSG ratio has been described to occur in situations in which the redox environment is more reducing, preventing oxidative modifications [56] and being associated with cell proliferation [54]. The proper coordination of cell proliferation is critical for the correct embryogenesis [57], and Nrf2 has been considered to control proliferation and differentiation by maintaining the redox state [58]. In addition, although not observed in this study, Thymol has been shown to increase GSH levels [59,60], which can affect cell proliferation in different ways, such as the regulation of c-Jun N-terminal kinase (JNK) and P38- mitogen-activated protein kinase (MAPK) pathways [61], modulation of cellular redox environment [62] and/or by affecting cytokine levels [63]. However, the relation between these effects and the modifications that may originate from the observed zebrafish eleutheroembryo malformations is not clear, and further research on this topic will be needed.

Nevertheless, glutathione is also involved in the cellular detoxification system, as it is used to conjugate a wide variety of exogenous compounds. In this context, phase II conjugation often involves glutathione-s-transferase (GST)-catalysed conjugation of GSH [64]. In the current study, GST activity increased following exposure to the highest concentration of Thymol. Although not described in aquatic species, Thymol has been shown to elevate GST activity in other non-target species [65,66,67,68], associated with a response to increased oxidative damage caused by reactive species. Yet, the increase in oxidative damage was not observed in the current study following Thymol exposure, as seen by the different oxidative-related parameters. Thus, other mechanisms might be involved. In view of this, GSTs are also implicated as modulators of cell proliferation and cell death by controlling the activity of members of the MAPK pathways, particularly by inhibiting JNK [69]. The inhibition of JNK has been shown to cause embryonic growth retardation, malformations and death of zebrafish [70,71], as observed in the current study. In accordance with this, Thymol has been suggested to modulate the in vitro expression of JNK [72,73] in a concentration dependent manner. Therefore, and although no in vivo information could be found in the literature, further studies are needed to elucidate the molecular mechanism involved in Thymol teratogenic effects.

Thymol exposure resulted in the disruption of behavioural responses in zebrafish eleutheroembryo, as observed by the lack of response to a threatening moving object and the increased distance moved in the dark period, which are associated with fear- and anxiety-like behaviours, respectively [74]. These emotional responses involve profound changes and specified activity patterns in the zebrafish brain [75]. Although a recent study has shown Thymol to improve the cholinergic nervous system and antioxidative stress in a cognitive dysfunction model [21], no supporting behavioural information could be found for zebrafish embryo. Yet, Thymol has been shown to affect the behaviour (depression- and antidepressant-like) of mice [76,77]. The modulation of emotional states in zebrafish are controlled by the habenula [78], an evolutionarily conserved structure of the vertebrate brain. The disruption of this structure was found to increase fear [79] and contribute to anxiety disorders [80], as observed following exposure to Thymol. The correct function and development of habenular circuits in zebrafish has a strong association with the correct embryo neurogenesis [81]. This is dependent upon complex intrinsic and extrinsic signalling factors interactions [82], with neurogenesis impairment being associated with various brain disorders. In accordance, altered neurogenesis has been previously described by changes in JNK signalling [83,84], and different behavioural phenotypes have been described in JNK-knockdown animal models [85]. Additionally, JNK has been shown as a dominant controller of behavioural moods [86]. Therefore, understanding the specific function of this signalling pathways in the Thymol-induced teratogenic effects will provide potentially important insights into the molecular mechanisms underlying the observed teratogenic effects.

## 5. Conclusions

In conclusion, the present study demonstrates the safety profile of both *E. arvense* and *M. tenuiflora* at sublethal concentrations during the early development of zebrafish. Yet, the data obtained further support the teratogenic potential of Thymol during early developmental stages as shown by the increased lethality and malformations. In addition, oxidative changes were observed, suggesting a change in the oxidative environment, which may be associated with effects at the proliferation level. While further studies are required to validate this hypothesis, the disruption of behavioural states further suggests alterations on the early embryonic signalling patterns. Taken together, the results obtained improve the risk assessment of these compounds, raising questions about the potential non-safe use of Thymol, which might have direct ecotoxicological consequences in non-target species.

## Figures and Tables

**Figure 1 toxics-09-00008-f001:**
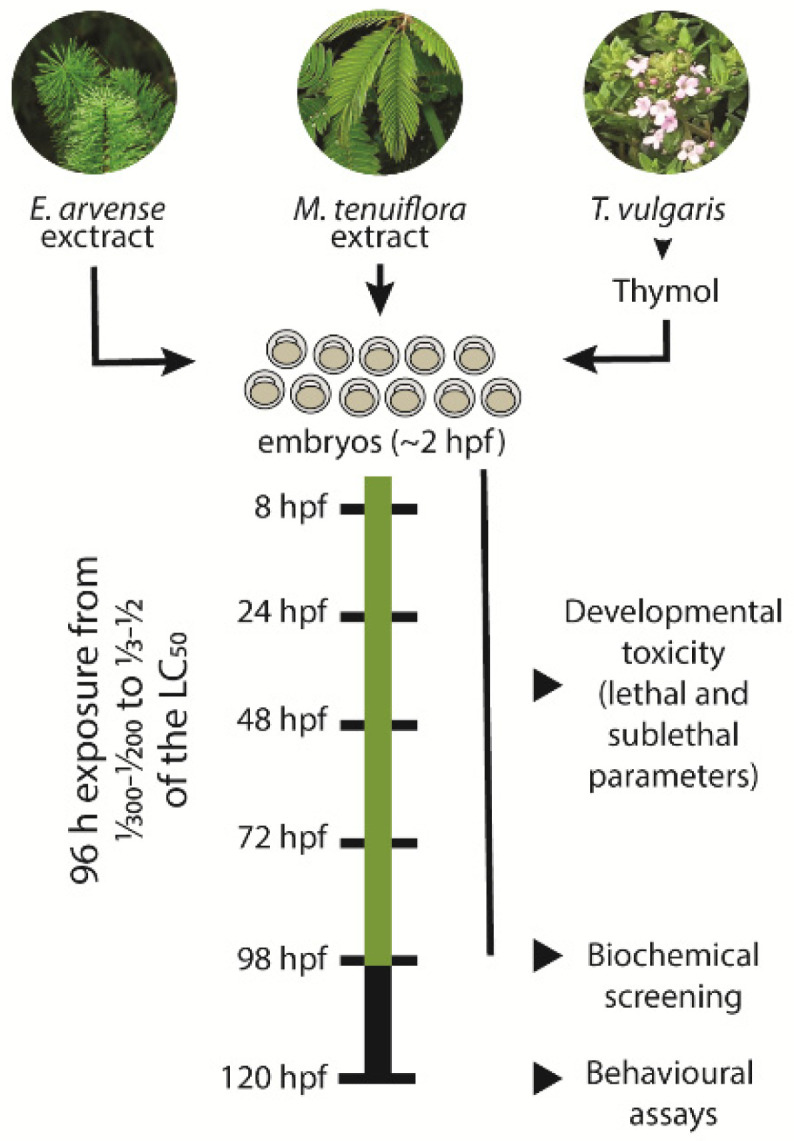
Schematic diagram of the zebrafish exposure to the plant-based fungicides. Collected embryos at around 2 h post-fertilization (hpf) were exposed to different concentrations of *Equisetum arvense* and *Mimosa tenuiflora* extracts and to Thymol for a period of 96 h. The selected concentrations varied from 1/300–1/200 to 1/3–1/2 of the determined LC_50_. During the exposure period, daily lethal and sublethal parameters were assessed. After 96 h exposure, eleutheroembryo were collected for biochemical screening of different biomarkers associated with oxidative stress, energetic metabolism, and neurotransmission. At 120 hpf, the locomotor activity of the eleutheroembryo was assessed using different behavioural paradigms.

**Figure 2 toxics-09-00008-f002:**
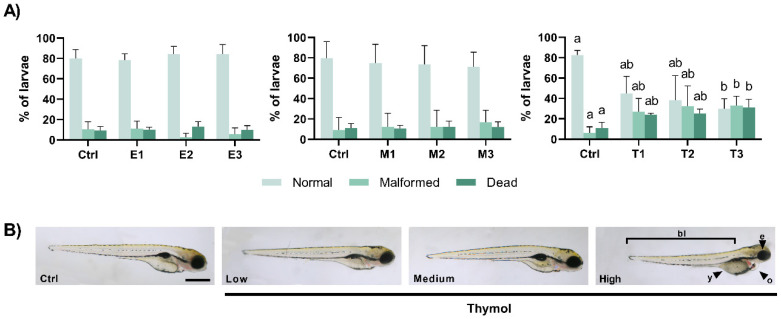
(**A**) Percentages of normal, malformed and dead eleutheroembryo at the end of the exposure period (at 98 hpf). Values are presented as mean ± SD of five replicates per treatment (*n* = 10 random embryos per replicate). Different lowercase letters represent statistical differences among treatment groups (one-way ANOVA, *p* < 0.05). (**B**) Representative views of the malformations observed in eleutheroembryo exposed to Thymol. Malformations were observed after exposure to the highest concentration of Thymol (T3) namely as abnormal eye (e), yolk (y), and pericardiac oedema (o) and by the decreased body length (bl). The scale bar represents 500 μm.

**Figure 3 toxics-09-00008-f003:**
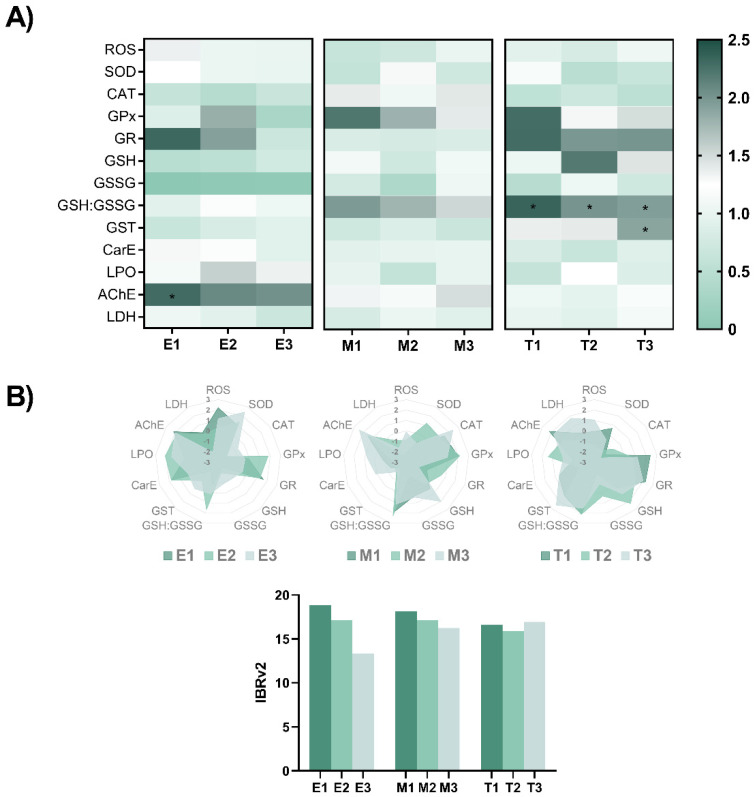
(**A**) Heatmap of biochemical parameters measured in zebrafish eleutheroembryo at the end of the exposure to the different phyto-fungicide. Data from at least five independent samples (*n* = 100 individuals per replicate). The data used for the evaluation of the biochemical parameters were normalised to the control group value. Parametric data is expressed as mean ± SD and statistical analysis was performed using one-way ANOVA followed by Tukey’s multiple-comparison test. The * indicate significant differences relative to the control group (*p* < 0.05). (**B**) Star plots of A values obtained and IBRv2 value for biomarker responses of zebrafish embryos exposed for 96-h to the different plant-based fungicides.

**Figure 4 toxics-09-00008-f004:**
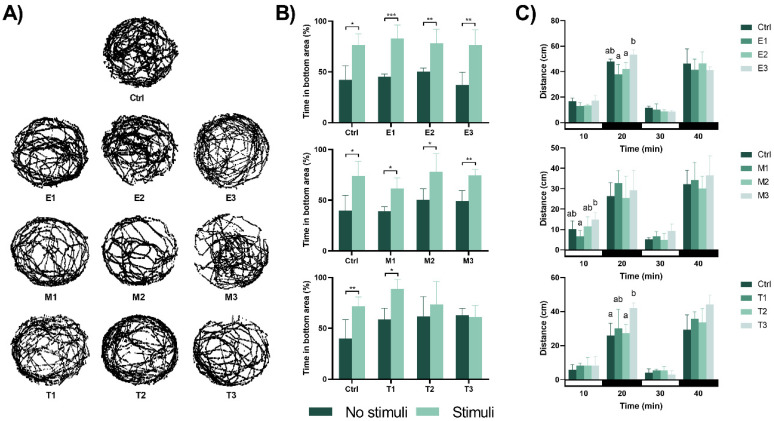
Effects of the different concentrations of the phyto-fungicide on zebrafish eleutheroembryo motor behaviour. (**A**) Representative swimming tracks of untreated and treated zebrafish eleutheroembryo at 120 hpf. No significant changes were observed between the different treatments and the control group during the 10 min recording period. (**B**) Avoidance behaviour of the zebrafish eleutheroembryo in the presence of an aversive stimulus. Data is expressed as mean ± SD from five independent replicates (5 eleutheroembryo assayed for each treatment). Statistical analysis was performed using *t*-test: * *p* < 0.05, ** *p* < 0.01 and *** *p* < 0.001. (**C**) Distance moved during the visual motor response test. Data represent the mean distance moved during 10 min assay and is expressed as mean ± SD from five independent replicates (5 eleutheroembryo assayed for each treatment). Statistical analysis was performed using one-way ANOVA followed by Tukey’s multiple-comparison test. Different lowercase letters indicate significant differences between groups (*p* < 0.05).

**Table 1 toxics-09-00008-t001:** Effects of the different exposures on spontaneous movements, heart rate and hatching rate in zebrafish embryos. Morphometric parameters were recorded in zebrafish eleutheroembryo at 98 hpf.

Phyto-fungicide	Group	24 hpf	48 hpf	72 hpf	98 hpf			
		Spontaneous Movements (units/min)	Heart Rate (beats/min)	Hatching Rate (%)	Body Length (mm)	Yolk Area (mm^2^)	Heart Area (mm^2^)	Eye Area (mm^2^)
*E. arvense*	Ctrl	2.0 (2.0–2.5)	123.6 ± 1.82	70.0 (65.5–71.5)	3.75 (3.71–3.77)	0.21 (0.20–0.21)	0.034 ± 0.001	0.082 (0.082–0.085)
	E1	2.0 (2.0–3.0)	124.2 ± 3.42	70.0 (66.0–71.0)	3.74 (3.72–3.78)	0.21 (0.21–0.22)	0.033 ± 0.003	0.083 (0.081–0.084)
	E2	1.0 (1.0–2.0)	117.0 ± 3.39	68.0 (63.5–71.5)	3.53 (3.49–3.65)	0.22 (0.22–0.22)	0.035 ± 0.003	0.075 (0.074–0.079)
	E3	2.0 (2.0–2.5)	121.2 ± 6.57	68.0 (61.5–68.5)	3.72 (3.65–3.80)	0.20 (0.20–0.21)	0.032 ± 0.004	0.081 (0.079–0.085)
*M. tenuiflora*	Ctrl	2.0 (0.5–2.0)	113.2 ± 7.19	62.5 ± 7.7	3.56 ± 0.20	0.22 (0.21–0.23)	0.038 ± 0.005	0.081 ± 0.015
	M1	1.0 (1.0–1.5)	115.0 ± 9.82	65.6 ± 5.3	3.48 ± 0.26	0.24 (0.23–0.24)	0.035 ± 0.006	0.086 ± 0.015
	M2	2.0 (1.0–2.0)	120.0 ± 5.92	56.4 ± 13	3.65 ± 0.14	0.24 (0.22–0.25)	0.039 ± 0.005	0.088 ± 0.010
	M3	1.0 (0.5–2.5)	118.8 ± 11.4	54.0 ± 16	3.60 ± 0.15	0.25 (0.22–0.25)	0.038 ± 0.005	0.087 ± 0.011
Thymol	Ctrl	2.0 (0.5–2.0)	144.2 ± 21.7	60.8 ± 7.6	3.47 ± 0.10 ^a^	0.21 (0.20–0.22) ^a^	0.031 ± 0.001 ^a^	0.079 (0.078–0.082) ^a^
	T1	0.0 (0.0–2.5)	159.6 ± 5.32	58.6 ± 3.1	3.44 ± 0.09 ^a,b^	0.21 (0.20–0.23) ^a^	0.037 ± 0.002 ^ab^	0.072 (0.061–0.079) ^a,b^
	T2	1.0 (0.0–1.0)	165.6 ± 5.68	59.4 ± 5.4	3.32 ± 0.05 ^b^	0.23 (0.22–0.24) ^a,b^	0.037 ± 0.006 ^a,b^	0.070 (0.063–0.073) ^a,b^
	T3	2.0 (1.5–2.0)	145.4 ± 20.5	59.0 ± 5.2	3.30 ± 0.04 ^b^	0.25 (0.24–0.28) ^b^	0.038 ± 0.003 ^b^	0.067 (0.062–0.073) ^b^

Parametric data is presented as mean and standard deviation while non-parametric data presented as median and interquartile range of five independent replicates. Statistical analysis was performed using the one-way ANOVA followed by Turkey’s test or by Kruskal–Wallis test followed by Dunn’s test (*p* < 0.05). Different lowercase letters represent statistical differences among groups for each analysed parameter.

## Data Availability

All data generated or analysed during this study are included in this published article (and its Appendix A files).

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
