# Peer review of "Teratogenic, Oxidative Stress and Behavioural Outcomes of Three Fungicides of Natural Origin (Equisetum arvense, Mimosa tenuiflora, Thymol) on Zebrafish (Danio rerio)"

_toxics, 2021, doi:10.3390/toxics9010008_

Round 1
Reviewer 1 Report
Developmental effects of three selected natural fungicides (E. arvense, M. tenuiflora, Thymol) to zebrafish (Danio rerio)
This works aims to assess the toxicological effects of commercially available herbal products containing Equisetum arvense, Mimosa tenuiflora, and Thymus vulgaris extracts to zebrafish embryos by assessing their effects on survival, development, oxidative stress and behavior parameters. The work is complete and well written. This is a very relevant issue given the increased use of plant-based extracts or compounds as alternative to synthetic fungicides allied to the lack of knowledge in this field. There are, however, some details that should be addressed to improve the manuscript.
Title: The title does not accurately describe the content of the manuscript as not only developmental effects were addressed; moreover, I would delete the word “selected”.
Abstract:
Line 21 – You mention “three alternative natural fungicides (E. arvense, M. tenuiflora, Thymol)”. For clarity, it should be mentioned that you tested the extracts of these plants have antifungal capacity, not the plants per se. A clear identification is given in lines 73-75. This should be made clear throughout the manuscript.
Line 27 – Instead of “in the highest Thymol-exposed group“ maybe you mean “in the organisms exposed to highest Thymol concentration”.
Line 31 – you mention “Notwithstanding, the results provide direct evidence of the teratogenic effects of Thymol, which might have consequences for the aquatic ecosystems.”, but in the discussion or conclusion you do not discuss or mention the possible consequences for the aquatic ecosystems, which I believe you should do. See my penultimate comment.
Introduction
Line 68 - Danio rerio needs to be italicized
Line 75 – You should explain what is thymol and which plant it is obtained from. You have some information in the Material and Methods section, but I think it should be referred in the Introduction. Also, it would be interesting to mention if any of these substances is currently used as fungicide. If so, that would increase the relevance of the present study.
Materials and Methods
Line 85 – “were prepared for E. arvense”
Line 91 - Danio rerio needs to be italicized
Line 96 – “nauplii”
Line 106 – “causes”
Line 109 – You mention “under the controlled conditions reported before.” But this is not clear concerning temperature – is it 28 ± 0.5 °C (line 86) or 28.5 °C (line 93)?
Line 114 – The sentence “This allowed the 96 h-LC50 and the 95 % confidence limits to be calculated as …” is not clear.
Line 115 – where is Figure S1? I did not find it nor in the manuscript nor in Appendix B. Is there any explanation for the CI to be large (e.g. 0.50-4.13 for the E. arvense extract. Could you repeat the tests with adjusted concentrations aiming to achieve a lower CI for the LC50 values?
Please note that there is no need to present the LC50 values in this section, as the text mentioned after these values in understandable even not mentioning the LC50 values. The LC50 values should be referred in the Results section, otherwise they might be unnoticed to some readers.
Lines 119-123 – why did you use different concentration ranges for the formulations/extracts?
Line 121 – LC50
Line 122 – mg L-1
line 129 – instead of “of the compounds to be tested” I suggest “of the test compounds”
Line 147 – you should present the complete speciues name of “T. vulgaris”
line 150 –You refer “During the early development of the embryo, daily lethal and sublethal parameters were assessed up to the end of the exposure.” This should be rewritten for clarity
line 159 – “centrifugation at 12 000g for 10 min…”
line 186 – “zebrafish larvae locomotor behavior”. Moreover, I would suggest to add an introductory sentence mentioning the different behavior assays carried out – this would prepare the reader for the several behavior tests described below.
Results
Line 218 – The subtitle “3.1. Thymol induces lethality and morphological abnormalities” should probably be replaced by a more generic one, reflecting the results presented in this subsection and not only the most relevant one.
Line 232 – caption of Table 1 should be rewritten – it is too generic in the current form. Also, replace “E arvense” by “E. arvense”. In addition, “M. tenuiflora” is not being shown correctly but this can be solved by formatting the Table.
Line 242 – you refer “while no differences were verified between other groups.” but this is not clear – do you mean that no increased mortality was observed for the other exposed groups?
Line 244 – both Figure 2A and 2B are required to support the sentence.
Line 260 – please consider rewriting this subtitle, following the same comment as for line 218. It is interesting that you label them as phyto-fungicides. Maybe you could introduce this word in the Introduction, rather than only in the Results section.
Line 263 – “Tables A2-A4” or, in alternative, change the tables nomenclature in the Supplementary Material file.
Line 269 – “increased GST activity” to conform with the nomenclature used throughout the manuscript (e.g. line 264). Also, “X2”
Line 272 – “E. arvense”
Line 289 – “Tables A5-A7” or, in alternative, change the tables nomenclature in the Supplementary Material file.
Line 307 – Since no significant effects of the tested formulations on zebrafish larva locomotor behavior was observed, I think there is no need to exhibit the part A of the figure in the manuscript. Thus, I would suggest to move part A of the figure to Supplementary Material file.
Line 309-310 – concerning the part A of the figure you refer that “No significant changes were observed between the different treatments and the control group during the 10 min recording period.” In tables A5 to A7 you show the statistical analyses supporting this sentence, but despite you mentioned that you also measured the percentage of time spent in each zone (line 191) you do not show the statistical analysis for this endpoint in the tables. This needs to be corrected.
The part C corresponds to “dark:light photoperiod stimulation”, but the correspondence with the “anxiety-like behavior” is not clear – this should be made clear to avoid any confusion.
Discussion
Line 325-335 – you should use the shortened form of the species names.
Line 330 – was
Line 333 – Results from the current study highlighted…
Line 334 – you mention that the 48h EC50 for Daphnia magna is 0.05-0.1 mg mL-1 [56]. However, the paper you cite is not clear concerning which species the EC50 refers to. By reading the Material and Methods section I believe this refers to Daphnia pulex. Anyway, to avoid any error I would suggest to write “in other aquatic organism (Daphnia sp.,…”. Also, to facilitate the comparison of this value to the LC50 value reported in the present study I recommend to use the same units, i.e., convert 0.05-0.1 mg mL-1 to mg L-1.
Line 337 – If the cited paper showed a 3x higher EC50 for Thyml you cannot say that “a previous study has shown that Thymol exhibits a higher toxicity”. As you know higher EC50 values stand for lower toxicity.
Lines 357 – 363 – this part of the discussion is based in results from other studies, not from the present study. I suggest either to delete or to rewrite this text.
Line 363 and 370 – You should uniformize the text concerning the word thymol – it is either Thymol (as in line 363) or thymol (as in line 370).
Line 375 – 382 – this text should be reduced – the discussion here presented is based on results from previous studies and, from my point of view, gives a very little contribution to understanding the observed thymol effects.
I would recommend to discuss the impact of these results – it is clear that the formulation containing thymol can have serious impacts to zebrafish. What will be their effects on the ecosystems? Does the bibliography suggests similar results to other aquatic species? Does this formulation represent a relevant concern for the aquatic biota? Are there any studies concerning its degradation/persistence in the aquatic environment? What further studies do you recommend to fully assess the ecotoxicological profile of thymol? Given that zebrafish is often used as a model for vertebrates does this suggest potential toxic effects to other vertebrates including humans? How do the LC50 values reported in the present study compare to the LC50 values of other natural or synthetic fungicides?
I would also recommend to decrease the number of references, whenever possible.
References
References should be reviewed. For instance, in ref. 37, you should replace “(danio rerio)” by “(Danio rerio)”

Author Response
Responses to the comments of Reviewer #1
- Title: The title does not accurately describe the content of the manuscript as not only developmental effects were addressed; moreover, I would delete the word “selected”.
Response: According to the reviewer’s concern, the title was changed.
- Line 21 – You mention “three alternative natural fungicides (E. arvense, M. tenuiflora, Thymol)”. For clarity, it should be mentioned that you tested the extracts of these plants have antifungal capacity, not the plants per se. A clear identification is given in lines 73-75. This should be made clear throughout the manuscript.
Response: The authors apologize for this confusing sentence. The manuscript was revised accordingly.
- Line 27 – Instead of “in the highest Thymol-exposed group“ maybe you mean “in the organisms exposed to highest Thymol concentration”.
Response: We acknowledge the suggestion made by the reviewer and changes were made accordingly.
- Line 31 – you mention “Notwithstanding, the results provide direct evidence of the teratogenic effects of Thymol, which might have consequences for the aquatic ecosystems.”, but in the discussion or conclusion you do not discuss or mention the possible consequences for the aquatic ecosystems, which I believe you should do. See my penultimate comment
Response: The discussion was changed to include these ideas.
- Line 68 - Danio rerio needs to be italicized
Response: Latin words were italicized in the revised version of the manuscript.
- Line 75 – You should explain what is thymol and which plant it is obtained from. You have some information in the Material and Methods section, but I think it should be referred in the Introduction. Also, it would be interesting to mention if any of these substances is currently used as fungicide. If so, that would increase the relevance of the present study.
Response: The authors apologize for not introducing the referred information. Changes were made in the introduction to clarify these faults. Relative to the use of these compounds as fungicides, there are some products available on the organic farming market under different names such as the ones used for this study. Yet, although there are plenty of laboratory studies on their antifungal effects, there is no available information, to our knowledge, about their application on crops. In the particular case of thymol, no commercial product was available at the time of the study and for that reason, extra pure thymol was used.
- Line 85 – “were prepared for E. arvense”.
Response: The authors apologize for not understating this question raised by the reviewer. In fact, the referred sentenced is relative to the preparation of stock solutions for all the compounds. As such, the plural was used.
- Line 91 - Danio rerio needs to be italicized
Response: Changes made accordingly.
- Line 96 - “nauplii”
Response: Changes made accordingly.
- Line 106 – “causes”
Response: The word was corrected.
- Line 109 – You mention “under the controlled conditions reported before.” But this is not clear concerning temperature – is it 28 ± 0.5 °C (line 86) or 28.5 °C (line 93)?
Response: The authors apologize for this typo. Changes were made to be consistent.
- Line 114 – The sentence “This allowed the 96 h-LC50 and the 95 % confidence limits to be calculated as …” is not clear.
Response: The sentence was rewritten.
- Line 115 – where is Figure S1? I did not find it nor in the manuscript nor in Appendix B. Is there any explanation for the CI to be large (e.g. 0.50-4.13 for the E. arvense extract. Could you repeat the tests with adjusted concentrations aiming to achieve a lower CI for the LC50 values? Please note that there is no need to present the LC50 values in this section, as the text mentioned after these values in understandable even not mentioning the LC50 values. The LC50 values should be referred in the Results section, otherwise they might be unnoticed to some readers.
Response: The authors apologize for this which might be associated to some problem in the MDPI platform as the supplementary materials were uploaded to the submission platform. The authors will have more careful and confirm that all files are in the platform. Regarding the confident intervals, in all tested compounds, the variation is explained by the narrow shape of the curve. In addition, the intension of the authors was not to show the LC50 in the results section as the purpose of its calculation was solely to define the concentrations for the subsequent experiments. As such, we kindly ask the reviewer to accept the manuscript in the current format.
- Lines 119-123 – why did you use different concentration ranges for the formulations/extracts?
Response: The authors acknowledge this question. In fact, for E. arvense, different concentrations were selected in order to facilitate sample preparation from the stock solutions.
- Line 121 – LC50
Response: The authors apologize for this typo.
- Line 122 – mg L-1.
Response: This typo was corrected.
- Line 129 – instead of “of the compounds to be tested” I suggest “of the test compounds”.
Response: The authors are grateful for the suggestion that was taken into account in the reviewed manuscript.
- Line 147 – you should present the complete speciues name of “T. vulgaris
Response: The complete name of the species was now included in the introduction of the manuscript.
- Line 150 –You refer “During the early development of the embryo, daily lethal and sublethal parameters were assessed up to the end of the exposure.” This should be rewritten for clarity.
Response: The sentence was now clarified.
- Line 159 – “centrifugation at 12 000g for 10 min…””.
Response: Changes were made accordingly.
- Line 186 – “zebrafish larvae locomotor behavior”. Moreover, I would suggest to add an introductory sentence mentioning the different behavior assays carried out – this would prepare the reader for the several behavior tests described below.
Response: The first sentence of the section was rewritten accordingly.
- Line 218 – The subtitle “3.1. Thymol induces lethality and morphological abnormalities” should probably be replaced by a more generic one, reflecting the results presented in this subsection and not only the most relevant one.
Response: The subtitle 3.1 was changed accordingly.
- Line 232 – caption of Table 1 should be rewritten – it is too generic in the current form. Also, replace “E arvense” by “E. arvense”. In addition, “M. tenuiflora” is not being shown correctly but this can be solved by formatting the Table.
Response: The referred caption was revised and E. arvense corrected. M. tenuiflora is shown correctly in the PDF generated by MDPI platform.
- Line 242 – you refer “while no differences were verified between other groups.” but this is not clear – do you mean that no increased mortality was observed for the other exposed groups?
Response: The sentence was clarified in the revised version of the manuscript.
- Line 244 – both Figure 2A and 2B are required to support the sentence.
Response: The reviewer is correct, and changes were made accordingly.
- Line 260 – please consider rewriting this subtitle, following the same comment as for line 218. It is interesting that you label them as phyto-fungicides. Maybe you could introduce this word in the Introduction, rather than only in the Results section.
Response: The subtitle was changed accordingly, and the word “phyto-fungicides” was included in the introduction.
- Line 263 – “Tables A2-A4” or, in alternative, change the tables nomenclature in the Supplementary Material file.
Response: The authors apologize for this typo which was changed accordingly.
- Line 269 – “increased GST activity” to conform with the nomenclature used throughout the manuscript (e.g. line 264). Also, “X2”.
Response: Changes were made accordingly.
- Line 272 – “E. arvense”
Response: The word was checked.
- Line 289 – “Tables A5-A7” or, in alternative, change the tables nomenclature in the Supplementary Material file.
Response: Tables were changed accordingly.
- Line 307 – Since no significant effects of the tested formulations on zebrafish larva locomotor behavior was observed, I think there is no need to exhibit the part A of the figure in the manuscript. Thus, I would suggest to move part A of the figure to Supplementary Material file.
Response: The authors appreciate the suggestion made by the reviewer. However, in order to show that the tested compounds did not cause any changes in the pattern of exploratory behaviour, the authors kindly ask the reviewer to accept the figure in the current format and to maintain the related results as a supplement to this figure.
- Line 309-310 – concerning the part A of the figure you refer that “No significant changes were observed between the different treatments and the control group during the 10 min recording period.” In tables A5 to A7 you show the statistical analyses supporting this sentence, but despite you mentioned that you also measured the percentage of time spent in each zone (line 191) you do not show the statistical analysis for this endpoint in the tables. This needs to be corrected. The part C corresponds to “dark:light photoperiod stimulation”, but the correspondence with the “anxiety-like behavior” is not clear – this should be made clear to avoid any confusion.
Response: The authors appreciate the issue raised by the reviewer. As for the time spent in each zone (upper and lower zones), this was just used to compare the time spent when the aversive stimulus (bouncing ball) was presented or not in order to infer the aversive-like behaviour and possible changes induced by the tested compounds. For this reason, the statistical analysis is not presented in the tables but in Figure 4B. As for the light/dark transition test, this is one of the most widely used tests to measure anxiety-like behaviour in zebrafish in which individuals with higher stress/anxiety levels show an increased locomotion activity (doi: 10.1016/j.ntt.2009.04.066 and 10.1016/j.neuro.2008.09.011). As these are standardized methods for the study of behavioural changes in zebrafish, the authors believe that no further explanation of the method is required to understand the results obtained and as such, the authors kindly ask the reviewer to accept the manuscript in the current state.
- Line 325-335 – you should use the shortened form of the species names.
Response: Changes were made accordingly.
- Line 330 – was
Response: Changes were made accordingly.
- Line 333 – Results from the current study highlighted…
Response: Changes were made accordingly.
- Line 334 – you mention that the 48h EC50 for Daphnia magna is 0.05-0.1 mg mL-1 [56]. However, the paper you cite is not clear concerning which species the EC50 refers to. By reading the Material and Methods section I believe this refers to Daphnia pulex. Anyway, to avoid any error I would suggest to write “in other aquatic organism (Daphnia sp.,…”. Also, to facilitate the comparison of this value to the LC50 value reported in the present study I recommend to use the same units, i.e., convert 0.05-0.1 mg mL-1 to mg L-1.
Response: The authors appreciate the suggestion made by the reviewer and changes were made in accordance.
- Line 337 – If the cited paper showed a 3x higher EC50 for Thyml you cannot say that “a previous study has shown that Thymol exhibits a higher toxicity”. As you know higher EC50 values stand for lower toxicity.
Response: The authors apologize for this error which was corrected in the revised version.
- Lines 357 – 363 – this part of the discussion is based in results from other studies, not from the present study. I suggest either to delete or to rewrite this text.
Response: This section of the discussion was rewritten.
- Lines 363 and 370 – You should uniformize the text concerning the word thymol – it is either Thymol (as in line 363) or thymol (as in line 370).
Response: The word “thymol” was uniformized throughout the text.
- Lines 363 Line 375 – 382 – this text should be reduced – the discussion here presented is based on results from previous studies and, from my point of view, gives a very little contribution to understanding the observed thymol effects.
Response: Although the discussion is based on the results from previous studies, the inclusion of these studies is required to raise hypothesis not only for the observed outcomes following exposure to Thymol but also to direct future studies. Therefore, the authors kindly ask to maintain the text in the current format.
- I would recommend to discuss the impact of these results – it is clear that the formulation containing thymol can have serious impacts to zebrafish. What will be their effects on the ecosystems? Does the bibliography suggests similar results to other aquatic species? Does this formulation represent a relevant concern for the aquatic biota? Are there any studies concerning its degradation/persistence in the aquatic environment? What further studies do you recommend to fully assess the ecotoxicological profile of thymol? Given that zebrafish is often used as a model for vertebrates does this suggest potential toxic effects to other vertebrates including humans? How do the LC50 values reported in the present study compare to the LC50 values of other natural or synthetic fungicides?
Response: Some of these ideas were included in the revised discussion. Yet, the authors would like to point that no other report has been found in the literature relative to the effects of Thymol in other aquatic species and, due to the issues raised particularly associated to the environmental fate of this compounds which remain elusive, some of the ideas presented in the previous discussion were removed. Regarding the possible studies to be done, some of them were described along the manuscript as the hypothesis were raised which would help to clarify the outcomes of the present study. Also, as referred in previous point 13, the calculation of the LC50 was used to define the concentrations to be used in the subsequent tasks. Yet, although the comparison to other species was now included in the discussion due to the lack of supporting information, several works show that pathway-based analysis of chemical effects in alternative models such as zebrafish could in fact predict therapeutic and harmful effects in higher vertebrates. Still, this remains one of the key challenges in pharmacological and toxicological research due to inter-species differences in metabolism, physiology, genetics and biochemistry. Therefore, in order to meet the reviewers' ideas, some changes were made in the manuscript.
- I would also recommend to decrease the number of references, whenever possible.
Response: The number of references was decreased whenever possible as suggested.
Reviewer 2 Report
The article entitled “developmental effect of three selected natural fungicides to zebrafish” presented to be published in Toxic describes some effects provoked by natural substances in aquatic organism. Modern agricultural habits has introduces natural substances with the aim of drive the activity to a more ecological one. Thus, the proper study of possible effects on the environment is crucial.
The results of the present study are publishable in Toxic if small changes are included.
The referee understands that Thymol substance is a derivative of Thymus vulgaris, however there is no reference to it. The other two are plant names Equisetum arvense, and Mimosa tenuiflora. Are the fungicide substances of the last two ones uknowns? I´ll add information in the introduction about the chemical properties of the substances of each plant derivative.
Finally,and as the authors conclude with the following sentence. “thymol migh have direct consequence from the aquatic environment”. Is there any data of those substances in aquatic environment? Are you using environmentally relevant concentrations? As far as the referee knows thymol could be photo-oxidized in water up to 50%. Which are the half lives of the selected subtances? Include this type of information in the introduction part.
MINOR CHANGES
1.-The first references to species names, must be long name in italics. Line 22. In the same way, in the title, the author used the long name of zebrafish, then why not for plants?
2.- Line 60: add the common name of Mimosa tenuiflora, as you did it for Equisetum arvense in the previous line
3.- line 68, Danio rerio in italics. Check the italics in the rest of the manuscript.
4.- Line 70. Be careful with following statement “low ethical constrains”, you are talking about a vertebrate so, there is no low ethical constrains for it. The referee assumes that the author is talking about the larvy, embryo status. However, the sentence must be change.
5.- The referee assumes that the extracts contains different ingredients which in a less concentration are also tested. Thus, even if the main compound is the plant, the microingridients added in the product are important. For instance, Matry product contains also Mn and Zn. Detailed information of the chemicals is needed. Line 81: it is mentioned “active ingredient”. Which one?
6.-Line 104.- the mention of the directive and legislations doesn´t ensure that the experiment and the methodological procedures are accepted by the competent administration. The reference of the accepted protocol must me added.
7.- Line 218. Change title of the subsection “thymol induces lethality and morphological changes”. There are not only presented the effects of thymol, so, please use a more general title. Same for line 286.
8.- Line 260. “Slighly” is not a scientific adjective, if it is supported by statistics, is fine, if it is because a tendency is seen then remove it. Change the title again to a more generalized one. “enzymes affected by exoposure” for instance.
9.- Table 1. Substance name Is not properly seen. The unite of the spontaneous movements parameter is not correct.
10.- Figure 2.- present the data either in tables (because of the numbers) or in graphs but not as both. It is repetitive. Include the data in the text or in the graph.
11.- figure 3.- “Data from at least 5 independent sampels (n=100/each)”. 2hat does it mean? N=5 or N=100 ?
Author Response
Responses to the comments of Reviewer #2
- The referee understands that Thymol substance is a derivative of Thymus vulgaris, however there is no reference to it. The other two are plant names Equisetum arvense, and Mimosa tenuiflora. Are the fungicide substances of the last two ones uknowns? I´ll add information in the introduction about the chemical properties of the substances of each plant derivative.
Response: According to the reviewer suggestion, this information was included in the introduction.
- Finally,and as the authors conclude with the following sentence. “thymol migh have direct consequence from the aquatic environment”. Is there any data of those substances in aquatic environment? Are you using environmentally relevant concentrations? As far as the referee knows thymol could be photo-oxidized in water up to 50%. Which are the half lives of the selected subtances? Include this type of information in the introduction part.
Response: The authors appreciate this comment. As no information could be found in the literature relative to the questions raised by the reviewer and knowing that some of these compounds might degrade before reaching the environment or become instable, the referred sentence was rewritten.
- The first references to species names, must be long name in italics. Line 22. In the same way, in the title, the author used the long name of zebrafish, then why not for plants?.
Response: According to the issue raised, changes were made.
- Line 60: add the common name of Mimosa tenuiflora, as you did it for Equisetum arvense in the previous line.
Response: The information was introduced as requested.
- Line 68, Danio rerio in italics. Check the italics in the rest of the manuscript.
Response: Manuscript was checked and species names and other Latin words were italicized.
- Line 70. Be careful with following statement “low ethical constrains”, you are talking about a vertebrate so, there is no low ethical constrains for it. The referee assumes that the author is talking about the larvy, embryo status. However, the sentence must be change.
Response: The authors apologize for the confusing sentence which was revised in accordance.
- The referee assumes that the extracts contains different ingredients which in a less concentration are also tested. Thus, even if the main compound is the plant, the microingridients added in the product are important. For instance, Matry product contains also Mn and Zn. Detailed information of the chemicals is needed. Line 81: it is mentioned “active ingredient”. Which one?
Response: The reviewer is right to point out that this extract contains other micronutrients which could were now included in the description of the reagents. Also, the description of the active ingredient is now
- Line 104.- the mention of the directive and legislations doesn´t ensure that the experiment and the methodological procedures are accepted by the competent administration. The reference of the accepted protocol must me added.
Response: As for the ethical statement required, from the legal point of view, prior to yolk depletion (before 5 days post fertilization), the embryos and larvae do not account as protected animals (doi: 10.1016/j.reprotox.2011.06.121) and as such, no ethical approval is required. Yet, the experiments were conducted by licensed researchers whose capabilities were recognized by the National competent authority (DGAV – Direção-Geral de Alimentação e Veterinária from Portugal) – at least FELASA B. This information is now added in the manuscript.
- Line 218. Change title of the subsection “thymol induces lethality and morphological changes”. There are not only presented the effects of thymol, so, please use a more general title. Same for line 286.
Response: As your suggestion, the titles have been changed.
- Line 260. “Slighly” is not a scientific adjective, if it is supported by statistics, is fine, if it is because a tendency is seen then remove it. Change the title again to a more generalized one. “enzymes affected by exoposure” for instance.
Response: The title of the subtopic was changed accordingly. However, as there are other parameters other than enzymes, the word “enzymes” was not used.
- Table 1. Substance name Is not properly seen. The unite of the spontaneous movements parameter is not correct.
Response: Regarding the issues raised, in the generated PDF, the table is in the correct format. As for the spontaneous movements, the unit was revised.
- Figure 2.- present the data either in tables (because of the numbers) or in graphs but not as both. It is repetitive. Include the data in the text or in the graph.
Response: The figure 2 was revised accordingly and changed to be presented as a bar graph.
- figure 3.- “Data from at least 5 independent sampels (n=100/each)”. 2hat does it mean? N=5 or N=100 ?
Response: For clarification, the sentence was revised.